# Degradation of Mechanical Properties of Graphene Oxide Concrete under Sulfate Attack and Freeze–Thaw Cycle Environment

**DOI:** 10.3390/ma16216949

**Published:** 2023-10-29

**Authors:** Ji Qian, Lin-Qiang Zhou, Xu Wang, Ji-Peng Yang

**Affiliations:** State Key Laboratory of Mountain Bridge and Tunnel Engineering, Chongqing Jiaotong University, Chongqing 400074, China; jiqian@cqjtu.edu.cn (J.Q.); 611220080025@mails.cqjtu.edu.cn (L.-Q.Z.);

**Keywords:** graphene oxide concrete, sulfate attack, freeze–thaw cycle, mechanical property, scanning electron microscope

## Abstract

In this paper, firstly, the effects of graphene oxide on the mechanical properties of concrete were investigated. Secondly, the degradation and mechanism of the mechanical properties of graphene oxide concrete (GOC) under sulfate attack and a freeze–thaw environment were investigated. In addition, the dynamic modulus of elasticity (MOE_dy_) and uniaxial compressive strength (UCS) of the GOC were measured under different environmental conditions. According to the test results, the incorporation of graphene oxide in appropriate admixtures could improve the mechanical properties of concrete in these two working environments. It is worth noting that this effect is most pronounced when 0.05 wt% graphene oxide is incorporated. In the sulfate attack environment, the MOE_dy_ and UTS of the GOC_0.05%_ specimen at 120 cycles decreased by 22.28% and 24.23%, respectively, compared with the normal concrete specimens. In the freeze–thaw environment, the MOE_dy_ and UTS of the GOC_0.05%_ specimen at 90 cycles decreased by 13.96% and 7.58%, respectively, compared with the normal concrete specimens. The scanning electron microscope (SEM) analysis showed that graphene oxide could adjust the aggregation state of cement hydration products and its own reaction with some cement hydration crystals to form strong covalent bonds, thereby improving and enhancing the microstructure density.

## 1. Introduction

Concrete is a versatile building material used for construction over the last few decades. It is made by mixing cement, water, and other materials. On the one hand, to achieve recycling and reuse, some academics have mixed waste materials (such as waste marble powder [1], coal gangue [2], and molybdenum tailings [3], etc.) from industrial processes into the fundamental components of concrete to protect the environment and meet required criteria. On the other hand, the addition of fibers to concrete is becoming increasingly popular to enhance the mechanical properties of the concrete. There are many different types of fibers available today, such as glass fiber [4], carbon fiber [5], basalt fiber [6], steel fiber [7], etc., that have been employed in a wide range of civil applications. Compared with the ordinary fiber, they have good integration with the construction and help in transferring the load as well as improving the physical and mechanical performance of the mixture materials [8]. The addition of all graphene and carbon fiber improves the properties of the concrete in several ways [9].

Cement is widely used in construction projects as the primary constituent of concrete and mortar. Nevertheless, the presence of brittle cracking defects in concrete and mortar results in a reduction in their ultimate compressive strength (UCS) and dynamic modulus of elasticity (MOE_dy_) as well as the loss of the internal alkaline environment [10,11]. Recently, the burgeoning field of nanomaterial research has prompted an increasing number of researchers to investigate methods to improve the performance characteristics (such as mechanical property, durability, and long-term performance) by incorporating nanomaterials [12,13]. The utilization of nanotechnology in construction presents a certain degree of complexity [14]. Specifically, carbonaceous nanomaterials, nanometals and metal oxides, and inorganic nanomaterials are the primary nanomaterials currently incorporated into concrete, with particular emphasis on graphene oxide as a noteworthy subject of investigation in the field of nanomaterials in recent times [15,16,17]. As shown in Figure 1, graphene oxide, as a derivative of graphene, exhibits both exceptional strength and flexibility as well as a substantial specific surface area. Furthermore, the presence of oxygen-containing functional groups on the surface of graphene oxide enables its dispersion in water and enhances its hydrophilicity [18]. This property has a significant impact on the formation of hydration products, as it facilitates the interweaving and penetration of these products into a uniform and dense microstructure. Consequently, this process leads to a significant reduction in the occurrence of internal defects, ultimately improving the strength, toughness, and durability of cement composites [19,20].

Lv et al. [21] investigated the effect of graphene oxide nanosheets on the microstructural and mechanical properties of cement composites. The results showed that the incorporation of graphene oxide nanosheets resulted in the modulation of flower-like crystal formation and a significant improvement in both tensile and flexural strength of the cement-based materials. Similarly, Indukuri et al. [22] used scanning electron microscopy (SEM) and X-ray diffraction (XRD) analysis to determine the properties of cement composites. Then, the influence of graphene oxide on the microstructure and reinforcing capacity of cement composites containing fly ash and silica fume was investigated. Jiang et al. [23] investigated the effect of incorporating polyvinyl alcohol (PVA) fibers and graphene oxide on the mechanical properties, durability, and microstructure of cement-based materials. The results showed that the incorporation of both PVA fibers and graphene oxide resulted in a significant improvement in the mechanical strength and durability of the cement-based materials, exceeding the performance of control specimens. In addition, Chintalapudi et al. [24] provided a comprehensive compilation of various studies that reported the enhancement of compressive strength in graphene oxide specimens. Gong et al. [25], Pan et al. [26], and Duan et al. [27] have also shown that the incorporation of graphene oxide in cementitious composites and mortars has a positive influence on their mechanical properties. The application of GO in concrete can improve the mechanical properties and durability of the concrete, and it can obtain better economic benefits under certain additive amount and use conditions [28,29]. Therefore, the application of GO in concrete has been widely paid attention to and studied, and it has been applied in many kinds of concrete structures and components, such as bridges, tunnels, subways, water conservancy projects, buildings, and other fields [30,31,32].

Sulfate attack has been identified as a prominent factor contributing to the reduced durability of concrete [33,34], with documented cases of damage due to sulfate attack in coastal regions. Degradation of the mechanical properties of concrete due to freeze–thaw damage in civil engineering practice in cold climates is recognized as the primary factor [35,36]. Therefore, the development of high-performance concrete is of great importance in various engineering applications. Yang et al. [37] used long-term immersion and dry–wet cycling as simulation methods to investigate the corrosion resistance of structures immersed in seawater or groundwater for long periods of time or structures frequently exposed to dry–wet processes. The results of their tests showed that the incorporation of graphene oxide in concrete can significantly improve its corrosion resistance coefficient. This improvement can be attributed to the presence of 3CaO·2SiO_2_·3H_2_O (abbreviated as C-S-H) within the internal structure of the specimens. Similarly, Cheng et al. [38] investigated the durability performance of concrete specimens exposed to both sulfate attack and dry–wet cycles. The results of their study demonstrated that the recently established integral area of sulfate ion distribution effectively served as an appropriate indicator for characterizing the non-uniform degradation patterns observed in sulfate-attacked concrete. In addition, they presented a novel approach, which is based on homogenization theory, for predicting the extent of deterioration in compromised concrete structural elements. Mohammed et al. investigated the effect of graphene oxide on concrete properties through tests that showed that graphene oxide had improved resistance to chloride ion attack and to water permeability [39]; they also demonstrated that graphene oxide could refine the pore structure of cementitious materials, making them highly freeze–thaw resistant [40].

There is little research on the durability of GOC with respect to sulfate attack and freeze–thaw cycles. Because graphene oxide nanomaterials have a significant impact on the long-term mechanical properties and durability of structures in construction projects, it is imperative to investigate the degradation patterns and mechanisms that affect the mechanical properties of GOC in various corrosive environments. The adoption of such an approach is of immense importance in order to obtain a thorough understanding of the performance of this novel material and its use in the construction of structures. The primary objective of this work is to investigate the effect of graphene oxide nanomaterials on the mechanical properties of concrete, with a particular focus on the degradation of the mechanical properties of GOC in two corrosive environments (sulfate attack and freeze–thaw cycle) during its service life. The influence of different levels of graphene oxide incorporation on the mechanical properties of the specimens exposed to the environment was studied using SEM analysis of the microstructure of the concrete.

## 2. Materials and Experiment

### 2.1. GOC Material

The specimens were meticulously prepared according to the guidelines of GB/T 50081-2019 [41]. The dimensions of the specimens adhered to the standardized dimensions of 100 mm × 100 mm × 100 mm. The GOC specimens were composed of a combination of complex Portland cement, multilayer graphene oxide dispersion (from Suzhou Tanfeng Graphene Technology Co., Ltd., Suzhou China), aggregates (including crushed stone (from Chongqing Hexin Building Materials Co., Ltd., Chongqing, China) for coarse aggregates and manufactured sand (from Chongqing Hexin Building Materials Co., Ltd., Chongqing, China) for fine aggregates), and water.

(1)Complex Portland cement

All the indexes (fineness, stability, mechanical properties, etc.) of P.C. 42.5 (from Huaxin Cement Co., Ltd., Chongqing, China) complex Portland cement meet the requirements of GB175-2020 [42]. The compressive strength at 3 d and 28 d is not less than 15 MPa and 42.5 MPa, respectively. Its chemical composition is shown in Table 1.

(2)Multilayer graphene oxide dispersion

The multilayer graphene oxide was subjected to freezing, drying, and dispersing using the modified *Hummers* method [43]. The resulting product showed no precipitation and easy dispersion. The dispersion of graphene oxide appeared as a black oily liquid with a concentration of 10 mg/mL. The parameters and components of graphene oxide are shown in Table 2.

The microstructures of the graphene oxide sheet were investigated using scanning electron microscopy (SEM) (from Thermo Fisher Scientifi, Waltham, MA, USA) and transmission electron microscopy (TEM) (from Thermo Fisher Scientifi, Waltham, MA, USA), as shown in Figure 2. Figure 2a shows a thin sheet, revealing the substantial specific surface area possessed by graphene oxide. Figure 2b shows the presence of numerous folds on the surface of the graphene oxide sheet.

(3)Concrete aggregates

The GOC specimens use a coarse aggregate consisting of crushed stone graded 5–20 mm, while the fine aggregate consists of mechanism sand with the specifications shown in Table 3.

### 2.2. Experiment

It should be noted that the specimens were cured at a pool with a constant temperature around 20 ± 1 °C for 28 days, and were taken out 15 min before the test. The main procedure of the test in this paper is shown in Figure 3.

#### 2.2.1. Specimen Preparation

The GOC specimens used in the test were prepared with a water–cement ratio of 0.50 and a sand content of 35%. Before the test, graphene oxide dispersions with different mass ratios (0, 0.02, 0.05, 0.08, 0.11, 0.14, and 0.17 wt%) were first mixed with water. The concrete mixture was prepared according to the proportions shown in Table 4 and then poured into triple 100 mm × 100 mm × 100 mm molds. Subsequently, the marked specimens were placed in a curing room and subjected to standard conditions for the specified time. All procedures followed the guidelines specified in GB/T50082-2019 [44].

#### 2.2.2. Test Situation

(1)Basic test

For the basic test, the parameter of interest was the incorporation of graphene oxide at levels of 0, 0.02, 0.05, 0.08, 0.11, 0.14, and 0.17 wt%. The curing time of the specimens was set at 28 days. The GOC_0.02%_ indicates that the specimen contains 0.02 wt% graphene oxide.

(2)Sulfate attack test

The sulfate attack test used a Na_2_SO_4_ solution with 10 wt% concentration. The specimens were subjected to a series of dry–wet cycles, specifically 0, 30, 60, 90, and 120 cycles, as shown in Table 5. SA_30_-GOC_0.02%_ indicates a GOC specimen with 30 wet-dry cycles and 0.02 wt% graphene oxide.

(3)Freeze–thaw cycle test

The specimens were subjected to a series of freeze–thaw cycles, specifically 0, 30, 60, and 90 cycles, as shown in Table 6. FtC_30_-GOC_0.02%_ indicates a GOC specimen with 30 freeze–thaw cycles and 0.02 wt% graphene oxide.

#### 2.2.3. Measurement Indexes

The test was performed according to the guidelines of GB/T 50081-2019 [26] and included the measurement of the MOE_dy_ and UCS and SEM of the specimens. The detailed steps are as follows:
Once the number of cycles reached a predetermined test design, the surface moisture of the removed specimen was dried to assess surface damage.The resonance method was used to measure the MOE_dy_ of the specimen. A layer of petroleum jelly was applied as a coupling medium on the test surface of the specimen, which was positioned in the center of the polystyrene plate. Then, the excitation transducer rod of the DT-20 device (from Tianjin Yida Experimental Instrument Factory, Tianjin, China) was gently pressed at 1/2 of the center line of the test surface, while the receiving transducer rod was gently pressed at a distance of 5 mm from the end of the center line of the test surface. The test results obtained were calculated and processed according to Equation (1),
(1)$$E_{a}=13.244\times 10^{-4}\times WL^{3}f^{2}/a^{4},$$
where *E_a_* is the MOE_dy_ of the specimen (MPa); *a* is the cross-sectional length (mm); *L* is the length (mm); *W* is the mass (kg); and *f* is the vibration frequency of the fundamental frequency (Hz).The HUT-1000 device (from Jinan Sanqin Testing Technology Co., Ltd., Jinan, China) was used to measure the UCS of specimens. The specimen was placed in the center of the lower platen of this device and pressing was started. The loading speed was 1.2 ± 0.2 kN/s. When the load became sharply smaller and the specimen deformation sharply increased, the setup automatically controlled to stop the loading and the data were recorded when the specimen was destroyed. The test results were calculated and processed according to the following Equation (2),
(2)$$f_{{}^{150}}=0.95f_{{}^{100}},$$
where *f*_100_ is the UCS of the test specimen of 100 mm ×100 mm × 100 mm and *f*_150_ is the UCS of the standard specimen of 150 mm × 150 mm × 150 mm.New fracture surfaces of the specimens were observed using the Quattro setup to investigate the effect of graphene oxide incorporation on the concrete microstructure at different cycle counts.

## 3. Results and Discussion

The mechanical properties of GOC with different levels of graphene oxide incorporation (0, 0.02, 0.05, 0.08, 0.11, 0.14, and 0.17 wt%) were investigated. It was observed that the MOE_dy_ and UCS of the GOC basically reached a peak at 0.08 wt% graphene oxide incorporation. After that, further increases in graphene oxide incorporation did not significantly improve the mechanical properties of the GOC. Then, the results of the mechanical properties of GOC with different levels of graphene oxide incorporation (0, 0.02, 0.05, and 0.08 wt%) were systematically investigated through sulfate attack and freeze–thaw cycle tests.

### 3.1. Results for Basic Test

The UCS of GOC with a curing age of 28 days showed a rapid and then slow increase with the incorporation of graphene oxide, as shown in Figure 4. The UCS essentially reached its peak value (51.86 MPa) at the incorporation of 0.08 wt% graphene oxide, which represents an increase of 14.79% compared with the normal concrete specimen (GOC_0%_). Thereafter, the UCS did not show a significant further increase with the increasing incorporation of graphene oxide, with all specimens showing an increase of approximately 15% compared with the normal concrete specimen.

Hardened cement paste is a solid-liquid-gas multiphase system consisting of CH (Ca(OH)_2_), C-S-H, ettringite (AFt), monosulfate (AFm), hydrated cement particles, micropores, and water or aqueous solution filled in the micropores. To explain the obtained results, four representative groups of specimens were selected for microstructural analysis. The specimens were designated as GOC_0%_, GOC_0.08%_, GOC_0.14%_, and GOC_0.17%_, and the SEM images are shown in Figure 5.

From Figure 5a, it can be clearly observed that there are a large number of C-S-H, AFt, and CH in the GOC_0%_ specimen, and it can also be observed that some cracks are distributed around the C-S-H. As shown in Figure 5b, the pore size of the GOC_0.08%_ specimen tends to be smaller and the number tends to be reduced, and its graphene oxide sheet and clusters inhibit the growth and diffusion of surrounding cracks to some extent [45]. Compared with the GOC_0%_ specimen, the cement hydration products of the GOC_0.08%_ specimen are more regular in morphology and arrangement. With a further increase in graphene oxide incorporation (GOC_0.14%_), as shown in Figure 5c, graphene oxide sheets and clusters appear in large numbers, and partial agglomeration or clustering [46] is already observed. Excessive graphene oxide instead affects the aggregation and stacking of the cement hydration products as shown in Figure 5d, which causes some pores in the concrete and a more pronounced distribution.

At present, some scholars [26] have shown that this adhesion is due to a strong covalent bond: such chemical reactions can create a strong covalent bond between the interface of the graphene oxide sheet and the cement matrix, which to some extent increases the integrity of the graphene oxide sheet as a skeletal lap between the cement hydration products. This increases the efficiency of load transfer from the cement matrix to the graphene oxide and thus improves the mechanical properties of the GOC.

In summary, at graphene oxide incorporation levels of 0–0.08 wt%, the graphene oxide can be well dispersed, participate in and regulate the cement hydration process, make the formation and aggregation of the cement hydration products more regular and effective, and reduce the cracks and pores inside the concrete [47,48]. When graphene oxide incorporation is too high, it is not well dispersed in the cement hydration process, which generates polymer nano-agglomerations. This change ultimately affects the gap between cement hydration crystals and aggregates, increases the porosity of the concrete, and does not improve the internal microstructural density of the concrete.

### 3.2. Results of Sulfate Attack Test

#### 3.2.1. Appearance Phenomena

The sulfate attack on concrete can be divided into two main categories: physical and chemical erosion [49]. During the physical attack, the cracking damage of concrete is caused by the swelling stress generated around the pore walls of the concrete, which is greater than the tensile strength; the swelling stress results from the pressure generated by the crystals on the pore walls. During the chemical attack, sulfate ions react with cement hydration products to produce swelling products that are about 2.5 times larger than the initial reaction phase, thus causing swelling cracking of the concrete, as shown in Figure 6.

This damage usually causes spalling, and many microcracks appear on the surface of specimens. The specimens show spalling at the edges and corners as the attack becomes more severe. As shown in Figure 7, the specimens maintained good integrity in the less severe case (60 cycles). In the case of severe erosion (120 cycles), the normal concrete specimens were severely damaged compared with the GOC specimens, which had relatively good integrity and less evidence of sulfate attack.

#### 3.2.2. Mechanical Properties

As shown in Figure 8, the MOE_dy_ and UCS of the GOC specimens under the sulfate attack test were both improved compared with the normal concrete specimens. With an increase in cycle number, the compressive indexes of the specimens showed a trend of “increases at first, and then decreases”, and the MOE_dy_ and UCS of the specimens reached the maximum at 60 cycles. The MOE_dy_ loss rate of the GOC specimens ranged from 22.282% to 28.252% (31.221% for normal concrete specimens), and the UCS loss rate ranged from 24.229% to 25.597% (31.397% for normal concrete specimens) after 120 cycles. With the increase in graphene oxide incorporation, the compressive indexes of the GOC specimens were all improved relative to normal concrete specimens, with the most significant improvement at 0.05 wt% graphene oxide incorporation, indicating that graphene oxide improved the concrete resistance to sulfate attack.

#### 3.2.3. SEM

To explain the results obtained in the previous section, the microstructure of specimens with different graphene oxide incorporation levels was compared and analyzed under 120 sulfate dry–wet cycles. The SEM images are shown in Figure 9. Figure 9a shows a large number of AFts produced by sulfate attack in normal concrete specimens. These AFt crystals are randomly arranged, and there are many cracks and pores around them, which further exacerbate the attack of sulfate ions on the concrete. After incorporating 0.02 wt% graphene oxide, the concrete microstructure was improved (Figure 9b), and cracks and pores tended to be refined and reduced. With a further increase in graphene oxide incorporation, the concrete microstructure was significantly improved (Figure 9c). Because many graphene oxide clusters are distributed in the pores as a filling and lapping effect, the pore area inside the concrete is effectively reduced, resulting in a significant decrease in the number of cracks and pores. When a large amount of graphene oxide is incorporated, graphene oxide sheets and clusters appear in large-scale agglomerations or clustering (Figure 9d). Many cement hydration crystals are distributed around them in a disorganized manner, the number of pores increases, and the density of the microstructure decreases, which instead affects the concrete resistance to sulfate attack.

The results show that graphene oxide can reduce the number of products generated by sulfate attack so that the microstructure of the concrete remains relatively dense when subjected to sulfate attack. On the other hand, it improves the erosion resistance of concrete by promoting the formation of regular cement hydration products and optimizing the crystal arrangement to reduce the size and number of microcracks and pores.

### 3.3. Results of the Freeze–Thaw Cycle Test

#### 3.3.1. Appearance Phenomena

Freeze–thaw damage in concrete is primarily caused by the freezing of water filling the capillary pores, which causes the concrete to expand. In turn, this forces the unfrozen water from the freezing area to flow around the capillary pores, creating a large migration pressure within the concrete. If the water content in the concrete capillary pore is too high, the expansion pressure on the pore wall will be significant, creating a large tensile stress around the pore. If this tensile stress exceeds the ultimate tensile strength of the concrete, internal microcracks will be generated, causing damage to the concrete structure.

The freeze–thaw damage in concrete is usually a combination of internal erosion and surface stripping, and the change in surface appearance is the most intuitive. Concrete surface damage includes water damage, large holes, pockmarks, cracks, slagging, and spalling. As shown in Figure 10, the surfaces of the specimens without freeze–thaw cycles were flat and free of obvious defects, with only a few small pockmarks and small holes. The surface of the normal concrete specimens showed more and more water stains, cracks, pitting, and spalling as the number of freeze–thaw cycles increased, while the GOC specimens (here, GOC_0.05%_ is taken as an example) showed less damage compared with them.

#### 3.3.2. Mechanical Properties

As shown in Figure 11, the compressive indexes of the specimens (MOE_dy_ and UCS) showed a trend of “first increase, and then decrease” as the number of cycles gradually increased, reaching a maximum at 30 freeze–thaw cycles. The compressive indexes of the GOC specimens were improved compared with the normal concrete specimens, and the compressive indexes improved more obviously with the increase in graphene oxide incorporation. After 90 freeze–thaw cycles, the MOE_dy_ loss rate of the GOC specimens ranged from 13.960% to 19.140% (21.730% for normal concrete specimens), and the UCS loss rate ranged from 7.590% to 11.350% (12.740% for normal concrete specimens). The results showed that graphene oxide improved the frost-resisting durability of concrete.

#### 3.3.3. SEM

To explain the results obtained in the previous section, the microstructure of specimens with different graphene oxide incorporation levels was compared and analyzed under 90 freeze–thaw cycles. The SEM images are shown in Figure 12. The typical damage of FtC_90_-GOC_0%_ specimens under repeated freeze–thaw cycles can be observed in Figure 12a, where wavy and fish-scale-type cracks are interconnected and tend to expand into a crack network. In addition, many cracks caused by freeze–thaw stress were distributed inside the concrete. The microstructure of the concrete was improved after incorporating a small amount of graphene oxide. As shown in Figure 12b, the graphene oxide sheets effectively stopped the interconnection, diffusion, and penetration of microcracks in the FtC_90_-GOC_0.02%_ specimens, but some cracks caused by freeze–thaw damage still appeared distributed on the surface of the cement matrix. With a further increase in graphene oxide incorporation, small-scale agglomeration or clustering occurred in the FtC_90_-GOC_0.05%_ specimens, as shown in Figure 12c, which enabled the concrete to maintain a relatively dense microstructure under repeated freeze–thaw cycles. Large-scale agglomeration or clustering was observed in the FtC_90_-GOC_0.08%_ specimen, as shown in Figure 12d. This led to a large increase in the number of pores distributed around the graphene oxide, which makes the effect of freeze–thaw damage more pronounced.

The results show that the number of defects in the concrete can be reduced. The microstructure is relatively dense, and strong covalent bonds with cement hydration products are formed by incorporating appropriate amounts of graphene oxide. This practice improves the performance of concrete against freeze–thaw damage and increases its freeze–thaw durability.

## 4. Conclusions and Future Directions

### 4.1. Conclusions

This paper investigates the degradation of the mechanical properties of GOC under sulfate attack and freeze–thaw cycles. Based on the mechanical behavior of graphene oxide cement slurry and the microstructure of graphene oxide concrete slurry observed using SEM, the main conclusions are as follows:(1)The mechanical properties of GOC specimens are higher than those of normal concrete specimens, showing a trend of “rapid and then slow increase,” in which the UCS of 0.08 wt% GOC reaches the maximum (51.86 MPa) for the first time, which increases by 14.79% compared with normal concrete specimens. SEM shows that this increase is related to the formation of strong covalent bonds.(2)For the sulfate attack environment, the performance of GOC shows a trend of “first increase, and then decrease” with the increase in the number of cycles. The addition of graphene oxide significantly improves the mechanical properties of concrete against sulfate attack, with the most significant improvement being achieved when 0.05 wt% graphene oxide is added. The MOE_dy_ and UTS of the SAGOC_0.05%_ specimen under 120 sulfate attack cycles decrease by 22.28% and 24.23%, respectively, compared with the normal concrete specimen (the decreases in MOE_dy_ and UTS of the other specimens range from 28.25% to 31.22% and 25.57% to 31.39%, respectively). SEM shows that graphene oxide can improve the microstructure of concrete in a sulfate attack environment, which is related to the cement matrix’s co-resistance to the degradation of the concrete’s mechanical properties due to swelling damage from corrosion products.(3)For the freeze–thaw cycle environment, the performance of GOC shows a trend of “first increase, and then decrease” with increasing cycle times. The resistance of concrete to mechanical degradation is significantly improved by the incorporation of graphene oxide, and this improvement is most significant at 0.05 wt% graphene oxide incorporation. The MOE_dy_ and UTS of the FtGOC_0.05%_ specimen under 90 freeze–thaw cycles decrease by 13.96% and 7.58%, respectively, compared with the normal concrete specimen (the decreases in MOE_dy_ and UTS of the other specimens range from 18.99% to 21.72% and 9.30% to 11.36%, respectively). SEM shows that this is related to the improvement of the concrete microstructure by graphene oxide and the inhibition of crack propagation and penetration under the freeze–thaw cycle.

### 4.2. Limitations and Future Directions

In this paper, the degradation study of the mechanical properties of GOC under a corrosive environment is still insufficient, and some problems are still to be solved, mainly in the following aspects:(1)The service environment of building structures is complex and variable. The experimental corrosion environment set up in this paper is only sulfate erosion or freeze–thaw cycle alone. Therefore, it is necessary to carry out a variety of environmental effects coupled erosion tests in subsequent research, such as the coupling erosion of chlorides and sulfates, the coupling of freeze–thaw and sulfate erosion, the coupling of chloride salt and freeze–thaw erosion, etc.(2)The erosion of building structures in the actual service environment is a long-term process. In this paper, only the degradation of the mechanical properties of GOC under a fixed number of corrosion times set in the laboratory is investigated. Therefore, long-term erosion tests need to be conducted in subsequent studies.(3)In this paper, only standard concrete specimens (100 mm × 100 mm × 100 mm) are used, and the mechanical properties of GOC members (compressive performance of columns, flexural performance of beams, etc.) need to be examined in subsequent studies.

## Figures and Tables

**Figure 1 materials-16-06949-f001:**
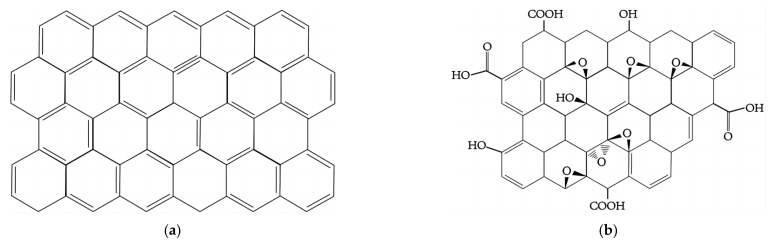
Molecular structure of graphene and graphene oxide. (**a**) Graphene; (**b**) Graphene oxide.

**Figure 2 materials-16-06949-f002:**
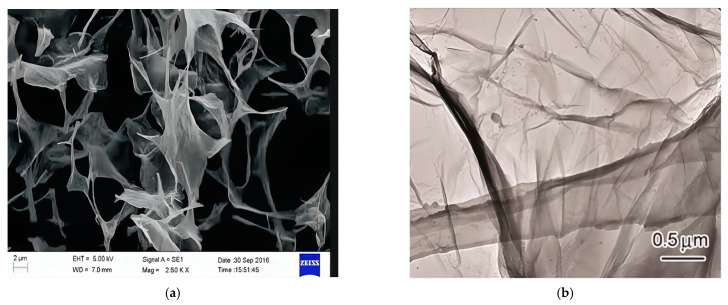
SEM and TEM images of the multilayer graphene oxide sheet. (**a**) SEM image; (**b**) TEM image.

**Figure 3 materials-16-06949-f003:**
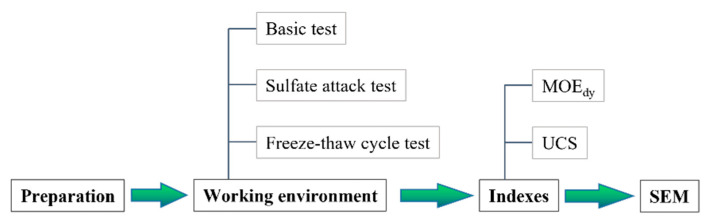
Testing process.

**Figure 4 materials-16-06949-f004:**
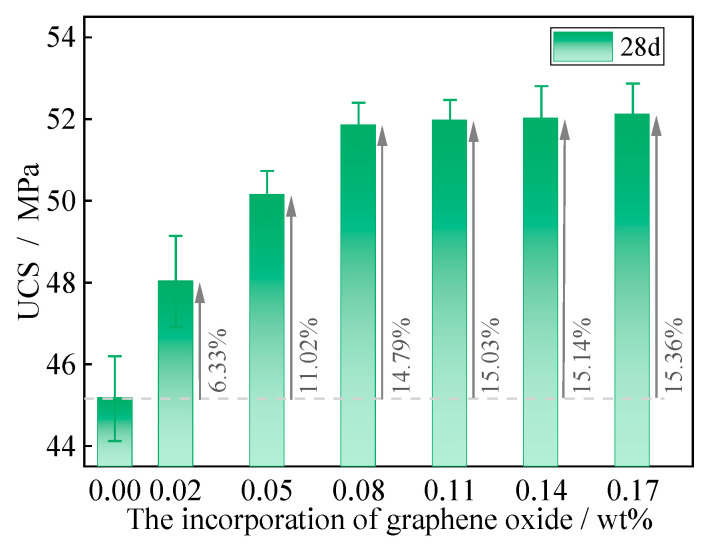
Mechanical property index of specimens under basic test.

**Figure 5 materials-16-06949-f005:**
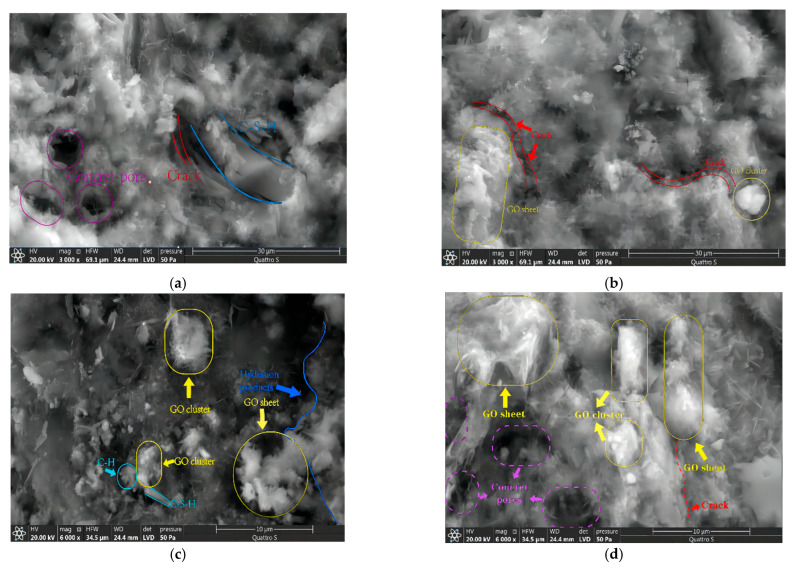
SEM images of specimens under the basic test. (**a**) GOC_0%_; (**b**) GOC_0.08%_; (**c**) GOC_0.14%_; (**d**) GOC_0.17%_.

**Figure 6 materials-16-06949-f006:**
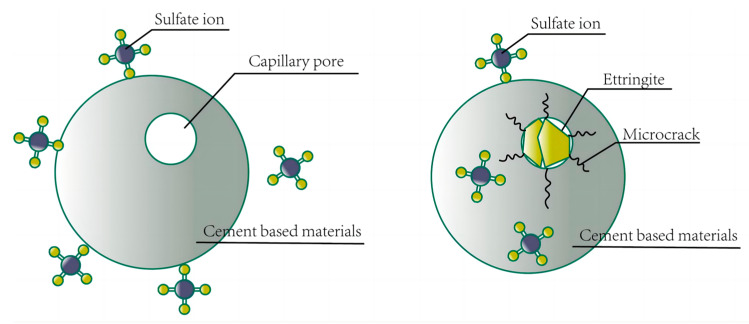
Schematic diagram of damage to concrete under sulfate attack.

**Figure 7 materials-16-06949-f007:**
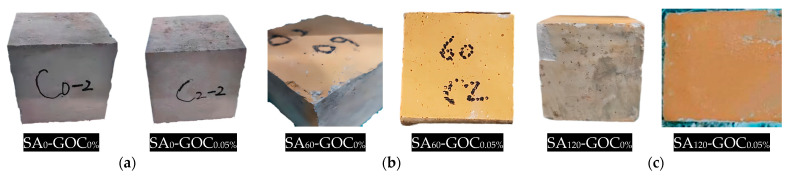
The appearance of specimens under the sulfate attack test. (**a**) 0 cycles; (**b**) 60 cycles; (**c**) 120 cycles.

**Figure 8 materials-16-06949-f008:**
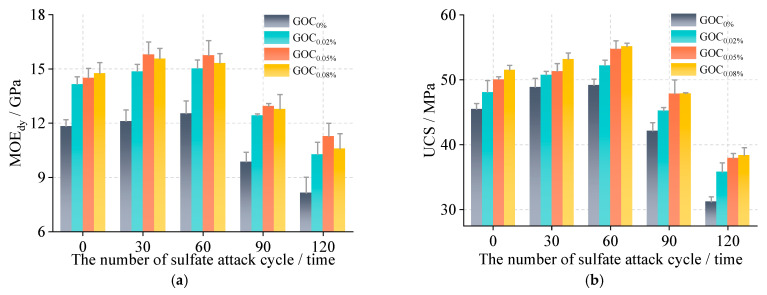
The mechanical properties of specimens under the sulfate attack test. (**a**) MOE_dy_; (**b**) UCS.

**Figure 9 materials-16-06949-f009:**
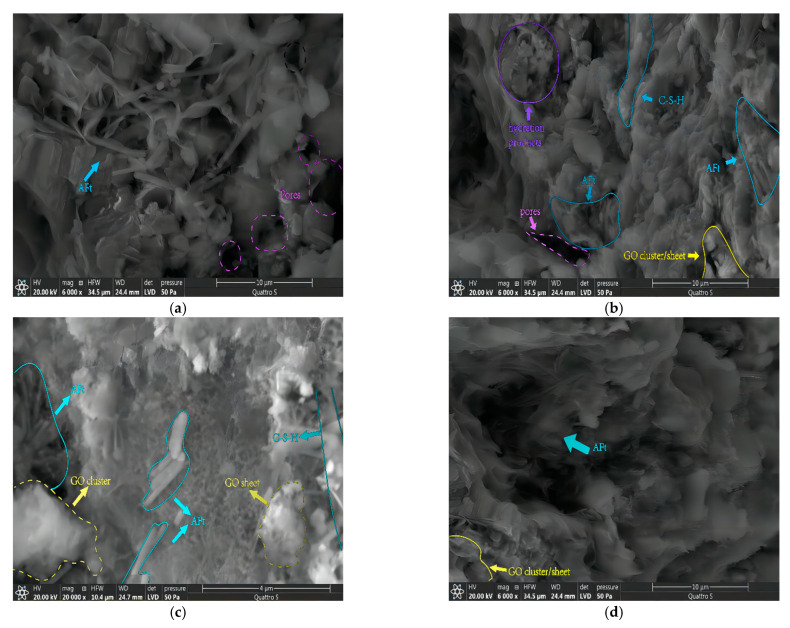
SEM images of specimens under the sulfate attack test. (**a**) SA_120_-GOC_0%_; (**b**) SA_120_-GOC_0.02%_; (**c**) SA_120_-GOC_0.05%_; (**d**) SA_120_-GOC_0.08%_.

**Figure 10 materials-16-06949-f010:**
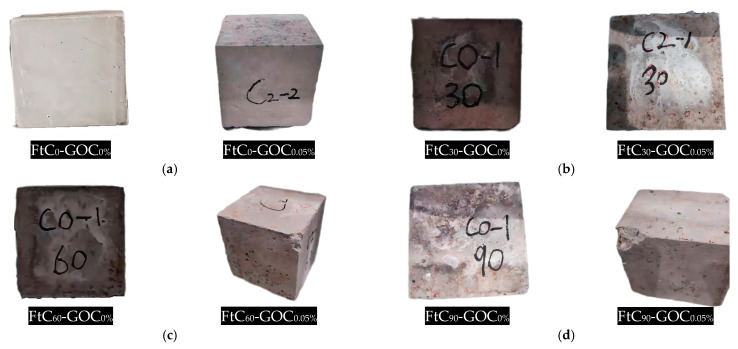
The appearance of specimens under the freeze–thaw cycle test. (**a**) 0 cycles; (**b**) 30 cycles; (**c**) 60 cycles; (**d**) 90 cycles.

**Figure 11 materials-16-06949-f011:**
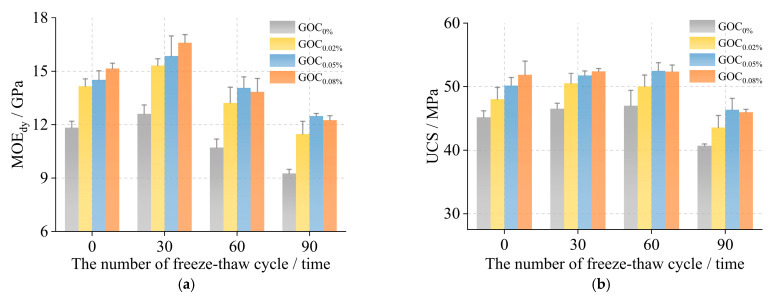
The mechanical properties of specimens under the freeze–thaw cycle test. (**a**) MOE_dy_; (**b**) UCS.

**Figure 12 materials-16-06949-f012:**
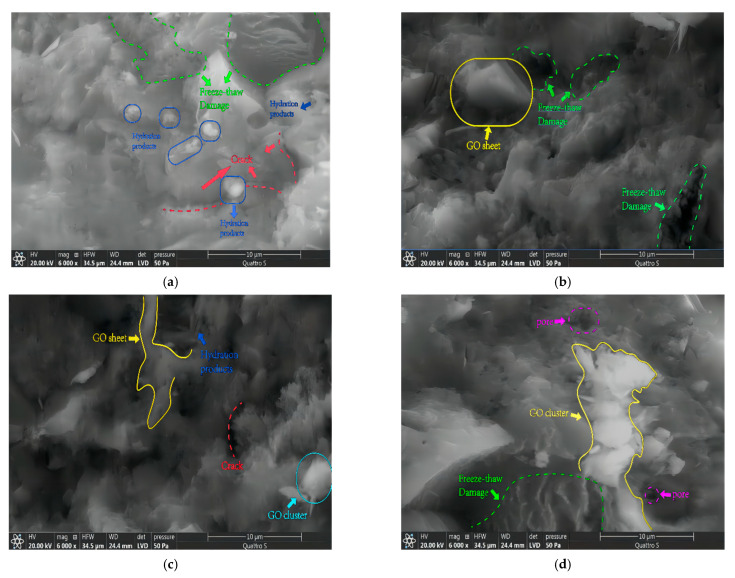
SEM images of specimens under the freeze–thaw cycle test. (**a**) FtC_90_-GOC_0%_; (**b**) FtC_90_-GOC_0.02%_; (**c**) FtC_90_-GOC_0.05%_; (**d**) FtC_90_-GOC_0.08%_.

**Table 1 materials-16-06949-t001:** The chemical components of the P.C. 42.5.

Components	CaO	SiO_2_	Al_2_O_3_	Fe_2_O_3_	MgO	K_2_O	SO_3_	Na_2_O	TiO_2_	P_2_O_5_	MnO	…
Percentage (%)	65.32	21.48	4.12	3.22	2.82	0.93	0.68	0.47	0.19	0.10	0.06	…

**Table 2 materials-16-06949-t002:** The physical parameters and chemical components of graphene oxide.

	Specification	Contents
Physical parameters	Purity	>95 wt%
	Thickness	3.42–7.82 nm
	Layers	0.79 nm
	Specific surface area	130–260 m^2^/g
	Lamellar diameter	12–40 μm
Chemical components	C	69.26%
	O	30.16%
	S	0.28%
	Si	0.16%
	Cl	0.11%

**Table 3 materials-16-06949-t003:** The particle gradation of fine aggregates.

Nominal diameter (mm)	0	0.15	0.3	0.6	1.18	2.38	4.72
Accumulated sieve residue (%)	100	89.34	74.98	52.96	37.45	18.76	0.96

**Table 4 materials-16-06949-t004:** Test design and mixture proportion (kg/m^3^).

Specimen	Cement	Manufactured Sand	Crushed Stone		Water	Graphene Oxide
			5–10 mm	10–20 mm		
GOC_0%_	398	623	450	734	195	0
GOC_0.02%_	398	623	450	734	195	0.08
GOC_0.05%_	398	623	450	734	195	0.199
GOC_0.08%_	398	623	450	734	195	0.318
GOC_0.11%_	398	623	450	734	195	0.437
GOC_0.14%_	398	623	450	734	195	0.556
GOC_0.17%_	398	623	450	734	195	0.675

**Table 5 materials-16-06949-t005:** The dry–wet cycle system for sulfate attack.

	Media Injection	Soaking	Media Discharge	Air Curing	Heating	Cooling
Initial temperature (°C)	27.0 ± 1.0	26.0 ± 1.0	26.0 ± 1.0	27.0 ± 1.0	27.0 ± 1.0	80.0 ± 5.0
Target temperature (°C)	26.0 ± 1.0	26.0 ± 1.0	27.0 ± 1.0	27.0 ± 1.0	80.0 ± 5.0	27.0 ± 1.0
Time (h)	0.25	17.5	0.25	0.5	3.5	2

**Table 6 materials-16-06949-t006:** The freeze–thaw cycle system for sulfate attack.

	Cooling	Freezing	Thawing
Initial temperature (°C)	20.0 ± 2.0	−19.0 ± 1.0	−19.0 ± 1.0
Target temperature (°C)	−19.0 ± 1.0	−19.0 ± 1.0	20.0 ± 2.0
Time (h)	0.5	2.0	1.5

## Data Availability

The data supporting the article’s findings are available from the corresponding author upon reasonable request.

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
