# Peer review of "Degradation of Mechanical Properties of Graphene Oxide Concrete under Sulfate Attack and Freeze–Thaw Cycle Environment"

_materials, 2023, doi:10.3390/ma16216949_

Round 1

Reviewer 2 Report

1.      “Error! Reference 32 source not found.” and “Lv et al. 12, Indukuri et al. 13, Jiang et al. 14, and many more” Fix this issue in whole manuscript and check all the citations once again (refer to Lv et al. [12], Indukuri et al. [13], Jiang et al. [14]).

2.      The abstract is informative, but it could be enhanced by discussing the main elements (background, materials and methodology, results, findings, contribution to the literature). Moreover, broader implications of the findings should be mentioned. Specifically, addressing how the results could impact the construction industry, graphene oxide concrete (GOC) practices, or other related fields would provide a more comprehensive perspective on the study's significance.

3.      In order to further enhance the comprehensiveness and applicability of this research, it is important  to include limitations and future orientations section before the conclusion. These future directions can serve as valuable guidance for subsequent studies in the field.

4.      The introduction provides a comprehensive overview of the importance of the work. The mentioned previous studies help to establish the context of the research. However, the authors should start explaining other types of concrete e.g. self-healing concrete, self-compacting concrete.. ect. Discuss these types before approaching the main topic for better understanding and differentiating. Refer to these latest references (discuss and cite them properly):

a.       https://doi.org/10.1016/j.jobe.2023.107527.

b.        https://doi.org/10.3390/pr10122745

c.      https://doi.org/10.1016/j.aej.2022.09.055

5.      It is advised to improve the conclusion and abstract by adding the quantitative results.

6.      The presentation of Table 4 lacks clarity and coherence.

Minor

Reviewer 3 Report

1. There are error messages in the text (links to images) - Error! Reference source not found

2. Some square brackets for literature are missing (from 19 to 33, 36)

3. Line 70 - The hydratation product C-S-H is correct like 3CaO.2SiO2⋅3H2O

4. Figure 7 c) According to the sample damage, the same sample is displayed

Round 2

Reviewer 2 Report

Good Work